# Instability Monitoring and Numerical Analysis of Typical Coal Mines in Southwest China Based on DS-InSAR

**DOI:** 10.3390/s22207811

**Published:** 2022-10-14

**Authors:** Maoqi Liu, Sichun Long, Wenhao Wu, Ping Liu, Liya Zhang, Chuanguang Zhu

**Affiliations:** 1School of Earth Science and Spatial Information Engineering, Hunan University of Science and Technology, Xiangtan 411201, China; 2College of Resource Environment and Safety Engineering, Hunan University of Science and Technology, Xiangtan 411201, China; 3Guizhou University Mining College, Mining College of Guizhou University, Guiyang 550025, China

**Keywords:** coal mining, DS-InSAR, deformation monitoring, numerical simulation, mountain stability

## Abstract

Most of the coal mines in Southwest China are located in mountainous areas with high vegetation coverage, and most activities are carried out under the mountains. The deformation monitoring and mechanical behavior analysis of the mining area helps reveal the typical mountain deformation and failure mechanism caused by underground mining activities and reduce the risk of mountain collapse in the mining area. In this manuscript, a research method for mountain stability in mining areas is proposed, which combines InSAR deformation monitoring with numerical analysis. Based on the high-precision deformation information obtained by DS-InSAR and the landslide range, a three-dimensional explicit finite difference numerical analysis method was used to reconstruct the landslide model. According to the layout of the coal mining working face, the variation mechanism of overlying stratum stress and the mountain slip in the coal mining process is inverted, and the mechanism of mountain failure and instability in the mining area is analysed. Based on the sentinel data, the experiment performed time series monitoring and inversion analysis of the mountain collapse in Nayong, Guizhou, China. The results show that mining activities a certain distance from the mountain will affect mountain stability, and there are specific mechanisms. From 2015 to 2017, the stress redistribution of overlying strata above the goaf area resulted in dense longitudinal cracks in the landslide body due to coal mining. The mountain is in a continuous damage state, and the supporting force to prevent collapse continues to decrease, resulting in a gradual decrease in landslide stability. Both the time series DS-InSAR monitoring results and numerical simulation results verify the actual occurrence and development of the on-site subsidence.

## 1. Introduction

The development and utilization of mineral resources are essential to support regional economic growth. However, it will also be accompanied by a series of ecological issues and geological disasters. For the southwest region, the landslide is one of the major geological disasters [1]. In recent decades, many researchers have studied how to control landslides and predict their temporal and spatial distribution to minimize landslide damage. Moreover, it is concluded that the factors that cause landslides include geology, landslide body shape, physics, and human activity [2,3]. Southwest China is one of China’s major coal mining areas. Taking Guizhou as an example, coal mining is mostly located in mountain areas due to Guizhou’s special topography and landforms. Therefore, mining impacts mountain stability in the mining areas, thus increasing disaster risks such as landslides. Moreover, other geological disasters threaten residents’ life and property safety in mining areas [4]. Scholars have conducted research on landslides caused by mining. Among them, the main research methods are physical simulation, numerical analysis, overlying rock failure model establishment based on actual monitoring data, and landslide mechanism analysis [5,6,7,8,9,10]. The mountain surface deformation is an important parameter for analysing mountain stability, and it is of great significance for large-scale detection, identification, long-term monitoring, and landslide prevention [11]. Traditional landslide monitoring methods include leveling instruments, GPS observation stations, crack meters, etc. However, the traditional monitoring method has a limited monitoring range, only monitoring points or lines, and is often limited by the influence of environment, time, etc. It cannot accurately reflect the overall change characteristics and landslide morphology trend. Therefore, further research is needed to monitor the surface deformation caused by coal mining [12].

Synthetic Aperture Radar Interferometry (InSAR) is a non-contact monitoring technology. It has the advantages of continuous spatial coverage, high precision, and no environmental restrictions [13]. Its differential interferometry mode is a mature air-to-ground measurement technology widely used in mine monitoring and can replace some traditional monitoring methods. Compared with traditional level monitoring, InSAR technology can provide a cost-effective monitoring method for mining areas, and there are many successful cases in practical applications. However, to achieve a higher precision monitoring level, it is necessary to mitigate the error sources, such as phase changes caused by atmospheric effects, residual noise, and temporal and spatial decorrelation [14,15,16,17,18,19]. InSAR time series analysis technology can solve the problems of temporal and spatial decorrelation and atmospheric delay in differential interferometry. A sufficient number of high-coherence points in time series analysis can make the obtained surface deformation parameter more accurate and reliable. Consequently, techniques such as Persistent Scatterers Interferometry (PSI), Small Baseline Subset (SABS), and Distributed Scatterers (DS) have been developed. Moreover, relevant scholars have proved that DS-InSAR has great advantages in mining areas because it can obtain a higher density of high-coherence points compared with PS-InSAR and SBAS-InSAR [20,21,22].

Similarly, DS-InSAR has been successful in ground deformation monitoring. For example, Reference [22] used DS-InSAR technology to process the HongQing (HQ) River coal mine in Ordos City. The obtained cumulative subsidence area was consistent with the coal mine spatial distribution. Reference [23] took ZhangShuangLou (ZSL) coal mine as an example, using DS-InSAR technology combined with the probability integration method to successfully invert the goaf position and probability integration parameters. Finally, the maximum relative error of the position parameters in the goaf was 16.1%, and the maximum relative error of the probability integral parameters was 26.67%. Reference [24] also took ZSL coal mine as an example, combining DS-InSAR and SBAS to monitor the coal mine surface deformation. Compared with the SBAS monitoring results, the former had more advantages than the latter in the surface deformation monitoring. Reference [25] used DS-InSAR in ShangYu District to obtain the surface subsidence parameters and compared them with the second-level data in this area. The results showed the reliability of this technology.

To sum up, the current research on mining areas using DS-InSAR technology mainly focuses on analysing surface subsidence and its impacts. Most scholars study the surface deformation of mining areas by monitoring accuracy, data processing, and combining various mathematical methods. However, few studies use this technology combined with numerical analysis methods, especially for studying mountain instability caused by surface deformation in mining areas. Based on this problem, this manuscript proposes combining the DS-InSAR technology and the numerical analysis method to study the mountain stability in the mining area to provide a reference for disaster prevention and control.

## 2. Materials and Methods

### 2.1. Overview of the Interested Area

The study area is located at the blue rectangle marked in Figure 1a and belongs to Nayong County, Bijie City, Guizhou Province, China. It is a small coal mine with a 300,000 t/a production capacity. The coal seam inclination angle of this mine is 7°, which belongs to the near-horizontal coal seam, and the inclined shaft development method is adopted. Figure 1b shows the study area overview, in which we can see the distribution of old kilns, goaf, and the coal mining face layout in the mine boundary. In the figure, except for the M14 and M16 mining areas, the other coal mining faces are arranged in the M10 coal seam, distributed at the mountain’s foot from far to near. M14 and M16 mining areas and 11,008 and 11,010 working faces have been mined, but 11,006 and 11,015 working faces have not been mined. Before the landslide occurred, mining work was carried out at the 11,013 working face. During the mining activity, much rock collapsed; therefore, the mining activity was stopped on 20 August 2017. Moreover, the landslide occurred on the eighth day after the mining work was stopped.

### 2.2. Sentinel Data and Processing in the Research Area

Terrain observation by progressive scans (TOPS) is a technology adopted by the Sentinel-1A satellite in wide swath imaging mode. The satellite beam swings from back to front along the azimuth direction during the flight to obtain a wider imaging area. The revisit period of Sentinel-1A is 12 days, and the theoretical upper limit of the surface subsidence rate monitored by the satellite is 42 cm/y. The image data captured in TOPS mode has a resolution of 2.7∼3.5 m in the range direction and 22 m in the azimuth direction [26]. The blue box indicated in Figure 1–shows the area selected for descending data of 36 scenes from 2015 to 2017. The temporal–spatial baseline is shown in Figure 2.

Due to the rapid changes in the Doppler centers’ azimuth of the adjacent strip images in TOPS mode, it is more sensitive to the azimuth registration error when the two images are subjected to differential interference. The azimuth registration error must be less than 0.001 pixels; otherwise, the interferogram phase will jump or be discontinuous. To meet the interference registration requirements of image data in TOPS mode, we use image registration to perform geometric registration under precise orbital conditions and then use enhanced spectral diversity technology to estimate the residual offset in the azimuth direction. Therefore, the azimuth registration accuracy is less than 0.001 pixels. This method is an effective way to eliminate phase jumps or discontinuities [26,27,28]. After completing the above registration, we performed differential interference with a single master, and DS-InSAR processing was carried out.Then, the interference pair is generated. Finally, the study area’s deformation rate was obtained using StaMPS (Stanford Method for Persistent Scatterers) for time series analysis. The research area of this manuscript is small, and the relative error of the high-coherence point in the vertical direction caused by the satellite orbit error is also small and can be ignored. The atmospheric error processing adopts PS technology, which has been considered in the DS-InSAR algorithm [29].

### 2.3. Methods

DS refers to scatterers with relatively consistent properties of ground objects. They can remain stable for a certain period and have a certain appearance and signal characteristics. The signal characteristics of DS provide a basis for the detection of DS points. The identification of homogeneous pixels is an essential part of the DS algorithm. For identifying homogeneous pixels, the current methods are mainly divided into non-parametric statistical assumptions and parametric statistical assumptions. Among them, non-parametric statistical assumptions mainly include the KS (Kolmogorov–Smirnov) test, the AD (Anderson–Darling) test, and the BWS (Baumgartner–Wei–Schindler) test. Furthermore, the parametric statistical hypothesis consists of the likelihood ratio test and the FaSHPS (fast statistically homogeneous pixel selection) test [30,31]. Each of the above test methods has its advantages. Since the research area of the paper is small, the computational efficiency can be ignored, considering the advantages of the KS test. Therefore, the homogeneous pixel identification in this manuscript adopts the KS test.

Statistically homogeneous pixels (SHP) are selected based on the amplitude data between two pixels, a threshold is set, and pixels more significant than the threshold are defined. Then, the SHP sample set is determined for estimating the covariance matrix. After the phase-linking (PL) process, the optimized phase value can be assessed by temporary coherence to remove the phase influence with poor optimization quality. Finally, the regional time series deformation can be estimated jointly with DS and PS. PL is a pivotal algorithm to realize image DS processing, and the phase triangulation algorithm (PTA) belongs to a method of PL, which is mainly reviewed in this section.

In interferometric processing, for example, given a set of N SAR images, one is selected as the master, and the remaining (N-1) images are properly re-sampled on the master grid. Assuming that **d** is a complex vector, there are: (1)$$d\left(P\right)={\left[d_{1}\left(P\right),d_{2}\left(P\right),\ldots d_{N}\left(P\right)\right]}^{T}$$

Among them, *T* represents the transposed vector, **P** is any image pixel, and $d_{i}$(*P*) is the complex reflectance value of the ith image of the data stack corresponding to pixel *P*. The backscattering energy is significant in PS. Moreover, the corresponding pixel contains the dominant scatterer, so **d** is an N-dimensional known vector. However, for DS, its pixels do not contain dominant scatterers, so **d** is an unknown and complex random vector. Given two data vectors, **d**(*P*1) and **d**(*P*2), at a certain level of significance, if the assumption that both *P*1 and *P*2 belong to the same probability distribution function holds, then the pixels *P*1 and *P*2 of the two images have statistical homogeneity. Ferretti et al. [30], 2011, proposed using the KS test to select statistically homogeneous pixels based on amplitude data, using the probability distribution function to determine whether the amplitude distribution of two pixels is consistent. Define the SHP sample set **Ω**, and **Ω** satisfies the matrix **Z**, $Z\in C^{n\times l}$, *n* represents the temporal domain, and *l* represents the spatial domain. Based on the central limit theorem, the matrix Z can be represented by a zero-mean n-variate complex circular Gaussian distribution. Therefore, for a complete statistical representation of DS, the SHP sample covariance matrix identified by the KS test or its normalized version, sample correlation matrix (SCM), can be estimated by the following formula: (2)$$C=\frac{ZZ^{H}}{\sqrt{{∥Z∥}^{2}{\left({∥Z∥}^{2}\right)}^{T}}}$$*H* indicates the Hermitian conjugation. In the PTA, it is proposed to decompose the SCM into two complex diagonal matrices and a full-rank real symmetric matrix: (3)Ψ=diag[Ψ]=diag[exp(jϕ)]
(4)$$\Sigma \left(P\right)=\Psi \Gamma \Psi^{H}$$
where ϕ is a superposition of the systematic consistency phase components, Σ denotes the complex coherence model of the SCM, and Γ is an N×N full-rank real symmetric matrix, $\Gamma \in R^{n\times l}$. The elements in the matrix represent the coherence values of all interferograms; Ψ is an N×N complex diagonal matrix whose elements represent the true phase values of pixel *P*. If the same set of phases can represent the set of SHP, the probability density function of SHP can be expressed as follows: (5)$$f\left(z\right)=\frac{1}{\sqrt{2\pi^{n}\det\left(\Sigma_{DS}\right)}}\exp\left\{-\frac{1}{2}{∥z∥}_{\Sigma_{DS}^{-1}}^{2}\right\}$$

Its second-order moment can be expressed as: (6)$$\Sigma_{DS}=\Psi \hat{\Gamma }\Psi^{H}$$

Calculate the maximum likelihood estimate of its true phase for the probability density function of the above formula, and set the phase of the first interferogram to be zero:(7)$$\begin{array}{cc} \hat{\phi } & ={\mathrm{argmax}}_{\phi }\left\{\prod_{i=1}^{l}\exp-\left(Z_{i}^{H}\Psi \Gamma^{-1}\Psi^{H}Z_{i}\right)\right\}\\ \; & ={\mathrm{argmin}}_{\phi }\left\{\mathrm{tr}\left[\Psi \Gamma^{-1}\Psi^{H}ZZ^{H}\right]\right\}\\ \; & ={\mathrm{argmin}}_{\phi }\left\{\psi^{H}\left(\Gamma^{-1}\circ C\right)\psi \right\} \end{array}$$

Here, the eigenvector corresponding to the smallest eigenvalue is considered the optimal phase. After obtaining the time series of DS points and phases, we combine PS and DS to perform a 2D linear regression analysis on the interference phase to remove DEM errors. Then, the residual phase caused by atmospheric delay, orbital error, etc., is removed using the different spatial-temporary characteristics. Finally, the deformation rate of the study area is calculated [32,33].

Based on obtaining the study area’s deformation rate and deformation range, the FLAC 3D finite difference numerical simulation method is used to simulate the landslide mountain in the study area, and the disaster occurrence process is inverted [34].

## 3. Results

### 3.1. Analysis of DS-InSAR Results

After data processing by the method described above, the annual average subsidence rate of the coal mine line of sight (LOS) surface in the study area from 8 January 2015 to 25 August 2017 was obtained. Since the incidence angle of the radar wave is known when the satellite transits, the subsidence rate from the LOS direction can be converted to the vertical direction through a simple geometric relationship. Figure 3 shows the results of the DS-InSAR experiment in the study area. Figure 3a shows the annual average deformation rate in the vertical direction of the study area. The white dotted box is the coal mine boundary. It can be seen from the figure that within the mine boundary, the surface and the top of the mountain have vertical downward displacement, and the maximum rate is −21 mm/a. Figure 3b is the contour map of the subsidence rate in the study area. The figure shows that the closer the mining area, the greater the subsidence rate. The left side of the red dotted line in the figure is the foot of the mountain, and the right side is the top of the mountain. For the mountain, referring to Figure 1b, it can be seen that the deformation rate near the coal mining face is higher than that in other parts of the mountain.

As shown in Figure 3a, we make a section of the mining area along $L_{1}$ and convert the deformation rate into displacement. Figure 4 shows that the study area presents an obvious funnel shape. The goaf in the figure is also the mining activity area in the earlier period. There was a large amount of surface subsidence in the early mining interval, and no landslide occurred in the mountain. However, the surface subsidence is slight in the mining activity area, and the landslide occurred in the mountain region. Comparing the periods, it can be shown that mining activities farther away from the mountain have less impact on the mountain stability, while mining activities near the mountain have more impact on stability.

### 3.2. Time Series DS-InSAR Results and Correlation Analysis

To further analyse the influence of underground mining activities on the mountain stability, we studied the relationship between the mining activities and the mountaintop displacement changes. The high coherence points in the mountaintop region were arbitrarily extracted for analysis according to the DS-InSAR time series results. Figure 5a shows the time series change diagram of the high coherence point displacement on the mountaintop. It can be seen from the figure that the extracted points have a certain change rule, and the changing trend is the same. The data of these points shown in Figure 5a are fitted. The fitting degree $R^{2}$ is 0.929, 0.925, 0.922, and 0.896, as shown in Figure 5b. From the fit value degree, it can be shown that the time change is highly correlated with the displacement change of the high coherence point on the mountaintop.

Generally, within a certain time frame, the longer the temporal baseline of the two scene images, the greater the deformation. However, in Figure 5b, the parts that conform to this rule are all on the right side of the master. For the left side of the master in the figure, from January 2015 to March 2016, the longer the temporal baseline, the smaller the displacement change of the high coherence point. Table 1 lists the combined coal mining time and image acquisition time, the mining location, time, and distance from the mountaintop. The table shows that between the first stage and the fifth stage during the coal mining process the relative position between the working face and the mountaintop was from far to near. By combining Figure 5 and Table 1, it can be shown again that mining at a position far from the top of the mountain has less impact on its stability. In contrast, mining has more impact on the mountain when it is closer to the mountain top region. The above analysis shows that coal mining in the study area affects mountain stability and has specific mechanisms.

Using StaMPS to process the DS-InSAR results, we obtain 35 time series change diagrams of displacement in the mining area. Due to space limitations, only the displacement time series changes on the right side of the master are shown, and two adjacent pictures with insignificant differences in results are removed. In Figure 6, the final results show that underground mining will cause surface subsidence, and the subsidence range and amount of the surface also increase with time. Since the working face gradually approached the mountaintop over time, the subsidence range was gradually approaching the mountain. Furthermore, for the right side of the master in Figure 5b, the displacement of the high coherence points on the mountain top increases with time. By combining the analysis of Figure 5b and Figure 6, it can be strongly demonstrated that underground mining affects mountain stability.

To verify the reliability of DS-InSAR experimental results conducting a field survey, Figure 7 shows the comparison between DS-InSAR results and the field, Figure 7b,c are the non-artificial steps found in the investigation at the mountaintop. The steps marked by dotted lines in the figures were obvious. By referring to relevant data and combining it with the experimental results, it is confirmed that the mountain sliding causes the steps. According to the experimental results of DS-InSAR combined with the numerical simulation method, the causes of mountain instability and disasters will be analysed.

## 4. Discussion

### 4.1. Mountain Stability Numerical Analysis Based on DS-InSAR

Based on the DS-InSAR experimental results combined with the relevant geological data, the research area’s numerical calculation model was established using three-dimensional finite-difference numerical simulation software. Figure 8 shows the sentinel satellite transit and numerical model. In order to reduce the number of numerical calculation units, improve the calculation efficiency, and increase the numerical simulation results’ reliability, we determined the model’s two-dimensional plane dimension according to the deformation range monitored by DS-InSAR in the study area. The vertical dimensions refer to the geological data. The Mohr–Coulomb elastic–plastic constitutive model is used for calculation [18], and the shear criterion and tensile yield criterion are as follows:
(8)$$\begin{array}{c} f_{s}=\sigma_{3}-\sigma_{1}N+2c\sqrt{N} \end{array}$$
(9)$$\begin{array}{c} f_{t}=\sigma_{t}-\sigma_{1} \end{array}$$
(10)$$\begin{array}{c} N=\frac{1-\sin\varphi }{1+\sin\varphi } \end{array}$$
where $\sigma_{1}$ and $\sigma_{3}$ are the maximum and minimum principal stresses, respectively; φ is the angle of friction; *c* is the cohesion; $\sigma_{t}$ is the tensile strength of the rock; and *N* is a parameter related to the angle of friction.

To fully understand the stability of the mountain under mining conditions, combined with the layout of the M10 coal seam working face, numerical calculations were performed at different positions from the mountaintop. Figure 9a–d show the results of these calculations. The red dots are the monitoring points of the summit displacement, and the remaining black dots A to D are the monitoring points during mining at different positions.

It can be seen from the above figures that when the working face is far away from the mountain, the mining influence range hardly affects the mountain. However, as the distance between the working face and the mountaintop decreases, the impact of underground mining will eventually affect the mountain.

At the same time as numerical calculation, the displacement monitoring was carried out at the corresponding positions, and the displacement monitoring results are shown in Figure 10. Figure 10a shows the displacement change of the top of the mountain when the mining position corresponding to Figure 9a–d is active. It is noticeable that from Position 1 to Position 4, the mountaintop displacement changes more obviously. Figure 10b(1–4) correspond to the mining situations at the corresponding positions in Figure 9a–d, respectively, and compare the surface displacement above the mining position and the displacement of the mountaintop.

Figure 10b(1) shows that the vertical downward displacement of the coal mining face corresponds to the surface monitoring point changes, but the change in the displacement of the mountain top is almost zero. It shows that mining at a position far from the mountain has little effect on the mountaintop. However, as shown in Figure 10b(1–4), the changing trend of the mountaintop displacement under the corresponding mining situation is gradually approaching the changing trend of the surface displacement of the mine site. Coal mining at the foot of the mountain and on the mountainside have significantly great impact on the mountain stability.

The coal mining design plan calls for leaving about a 10m thick coal pillar between working faces to protect the roadway. According to the mining sequence of the working face, we simulate the influence of coal mining on the mountain while excavating into the mountain. The results are shown in Figure 11.

The figure shows that when the working face excavates into the mountain, the impact range gradually widens. As shown in Figure 11c, when the working face excavates to the foot of the mountain, the mining activities affect the top of the mountain. Figure 11e,f show that when the mining activities are located under the mountain, they will have more impact on the mountain. It can be seen from Figure 11d–f that there is also a vertical downward displacement on the trailing edge of the summit. Here, comparing Figure 11d–f with the DS-InSAR results in Figure 7, we can conclude that the mountaintop step was formed by the sliding of the trailing edge caused by mining inside the mountain.

### 4.2. Analysis of Mountain Instability Mechanism in Mining Area

Figure 12 shows the stress change process during the excavation work advancing towards the mountain. The figure shows that the vertical downward stress of the coal wall gradually increases as the mining activities advance into the mountain, and the stress concentration is obvious. At the same time, from the compressive stress distribution in the vertical direction, the influence range of compressive stress change of overlying strata in the goaf gradually increases. Continuously, the underground mining activities will lead to unbalanced stress distribution in the surrounding rock of the goaf. Stress redistribution will constantly adjust this unbalanced state to restore a stress-balanced state. In the mining process, this cycle continues until the end of mining.

The rock mass in the original stress equilibrium state is affected due to the stress redistribution process to an equilibrium state. This kind of influence leads to its damage by shear stress or tensile stress as the overlying rock layer in the goaf produces fissures. The fissures’ development in the rock mass eventually forms the surface fissures of the rock mass, as shown in Figure 13a. The figure also shows that the instability and collapse of the mountain also spread along these cracks.

The cracks will weaken the rock body, and the rock body and its fissure will spread to the surface to form surface cracks. Under erosion and the penetration of rainwater, fissure water pressure will be generated. In a comprehensive analysis, whether the mountain (disaster-causing body, DCB) collapses is determined by four parameters: the supporting force $F_{s}$, the self-weight *M* of the collapsed body, the fissure water pressure $F_{w}$, and the rock mass damage degree *K*,where *K* is defined as the degree of rock mass damage under the continuous influence of external forces. When $F_{s}>F_{w}+M$, the DCB is in a stable state, and when $F_{s}=F_{w}+M$, the DCB is in a limited equilibrium state. When $F_{s}<F_{w}+M$, the DCB is unstable. The $F_{s}$ is determined by the rock body damage degree *K*. According to the above analysis, the larger the *K*, the smaller the $F_{s}$. Based on the analysis results of the DS-InSAR experiment and numerical simulation, from the first stage of mining activities to the fifth stage, the impact of mining on the mountain is positively correlated with time. It shows that the parameter *K* increases with time, and the $F_{s}$ is becoming smaller. When the supporting force is insufficient to support the DCB, it will fall and cause disasters.

### 4.3. DS-InSAR and Numerical Simulation

The value of the planar two-dimensional dimension of the model is usually manually set based on experience when we establish a numerical simulation model because there is typically no meaningful reference for this value. The established numerical model could be overly complex, lengthening the computation process. Additionally, the model could be made too tiny, requiring re-establishing it for analysis and resulting in erroneous results. The plane size of the model, however, can refer to the InSAR results because the results of InSAR monitoring can reflect a certain range. The determination of the two-dimensional scale of the numerical model corresponds to the DS-InSAR results. This is the same as the research in this publication. This essay also investigates the situation when mountain stability is impacted by coal extraction. Numerical simulation or DS-InSAR is utilized separately to produce unconvincing results, but their combination will increase the accuracy of the research findings. As seen from the analysis above, the results of the two approaches can be cross-verified. The research area is connected from the outside to the inside. As a result, the research methodology suggested in this manuscript can be employed by researchers with engineering backgrounds, making it possible to analyse engineering research more easily and come to more trustworthy conclusions.

The stability of the mountain was studied in this work using DS-InSAR technology in conjunction with a numerical analysis method. The mountain was modelled and inverted using FLAC 3D three-dimensional finite difference numerical simulation software. Surface deformation monitoring has shown the extensive application of the DS-InSAR technology with positive outcomes. The technology is also frequently used for surface monitoring in coal mines. Numerous geoscientists utilize FLA3D numerical simulation software, particularly in the mining sector. Because the two methods can be combined and utilized in similar technical contexts, the suggested method is thought to have strong stability.

## 5. Conclusions

In this manuscript, 36 scenes of Sentinel-1A descending data were acquired and processed by DS-InSAR, and a numerical analysis model was established based on the results. Simulation of the impact of coal seam mining on the mountain, combined with the comparative verification of the two results, allowed us to analyse the instability of the DCB. Research results are as follows:

The results of the DS-InSAR experiment show that coal mining influences mountain stability. There is a certain rule that the influence of mining activities on the mountain is positively correlated with time and negatively correlated with the distance between the mining face and the mountaintop.Combining the experimental results of DS-InSAR with the numerical simulation results can effectively explain the ground movement causes at the top and bottom of the mountain in the study area. Figure 7 shows the steps on the trailing edge of the mountaintop. This was caused by the downward displacement of the trailing edge of the mountaintop due to mining activities.According to the results of DS-InSAR and numerical simulation, the mining activities destroyed the mountain and reduced its stability. Therefore, we conclude that coal mining is one of the causes of mountain collapse.The experimental results show that the DCB collapses are determined by four parameters. Among these four relationships, the parameter *K* is significant in determining the relationship between $F_{s}$ and $F_{w}+M$.

## Figures and Tables

**Figure 1 sensors-22-07811-f001:**
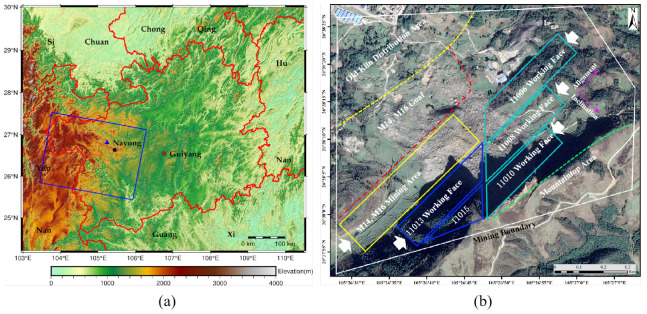
Geographical location and overview of the research area: (**a**) geographical location of study area; (**b**) overview of research area.

**Figure 2 sensors-22-07811-f002:**
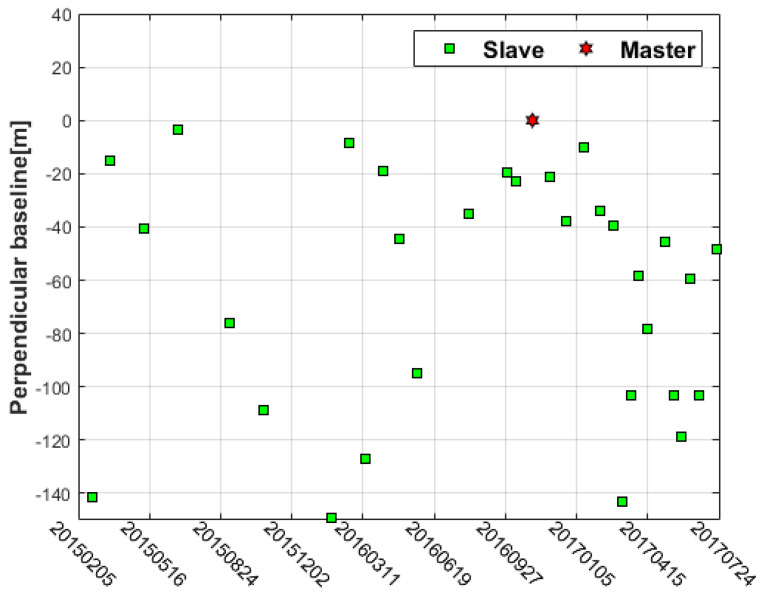
Temporal–spatial baseline.

**Figure 3 sensors-22-07811-f003:**
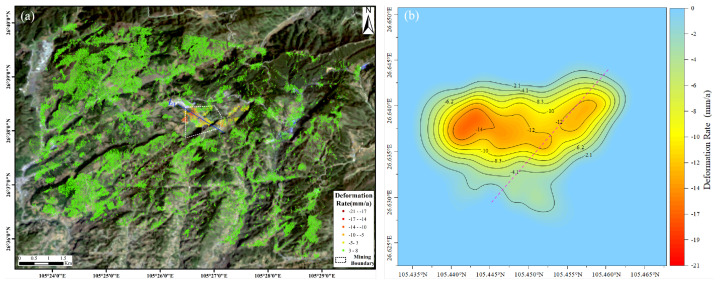
Result of DS-InSAR in the research area: (**a**) annual average deformation rate in the study area; (**b**) contour of deformation rate in study area.

**Figure 4 sensors-22-07811-f004:**
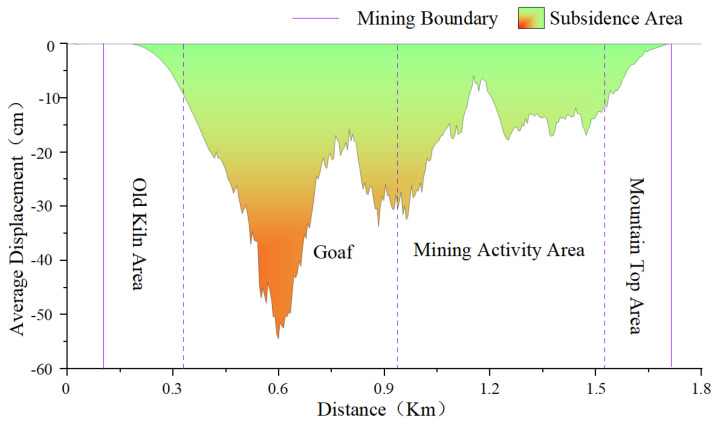
Trend profile of study area.

**Figure 5 sensors-22-07811-f005:**
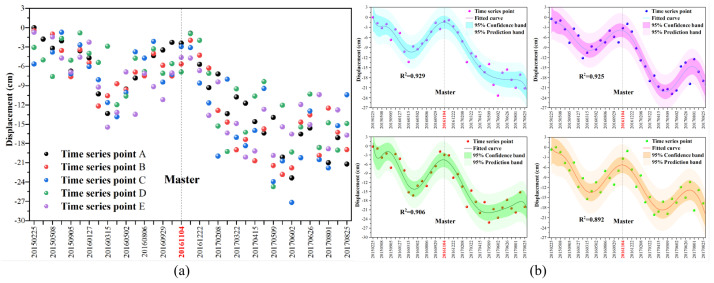
Variation of displacement time series of high coherence on the mountaintop: (**a**) displacement time series change; (**b**) displacement time series of fitted curve.

**Figure 6 sensors-22-07811-f006:**
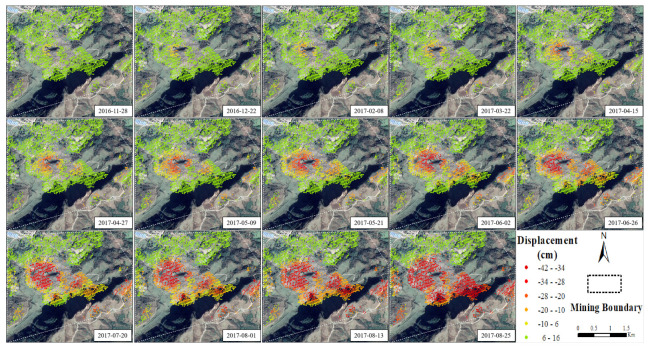
Time series change of mining area displacement in mining stage.

**Figure 7 sensors-22-07811-f007:**
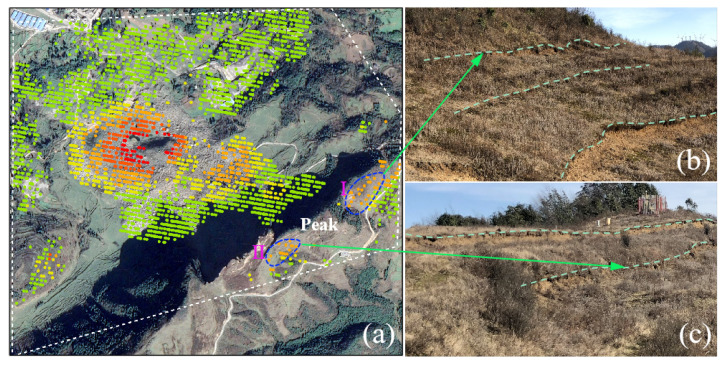
Comparison between DS-InSAR results with field investigation results: (**a**) DS-InSAR time series monitoring results; (**b**) Investigation area I; (**c**) Investigation area II.

**Figure 8 sensors-22-07811-f008:**
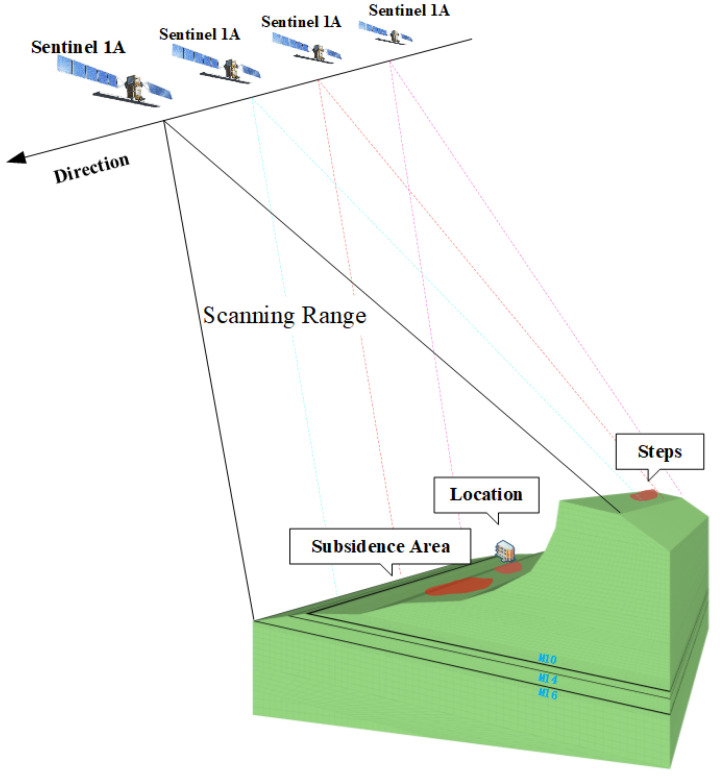
Schematic diagram of satellite shooting and numerical calculation model in mining area.

**Figure 9 sensors-22-07811-f009:**
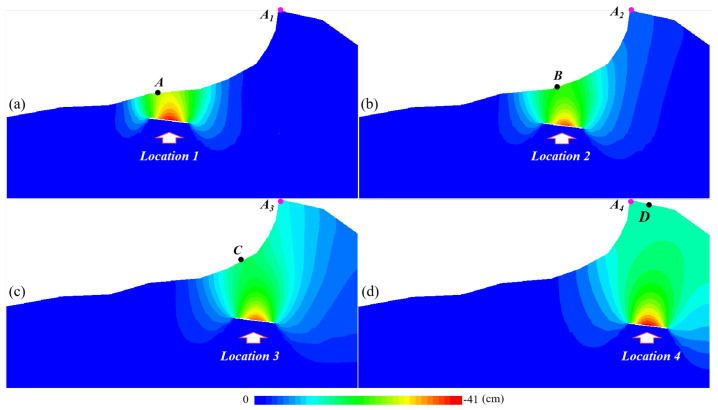
The influence of mining at different locations on the mountain: (**a**) simulation results of mining far away from the top of the mountain; (**b**) simulation results of mining at the foot of the mountain; (**c**) simulation results of mining under mountainside; (**d**) simulation results of mining under the top of the mountain.

**Figure 10 sensors-22-07811-f010:**
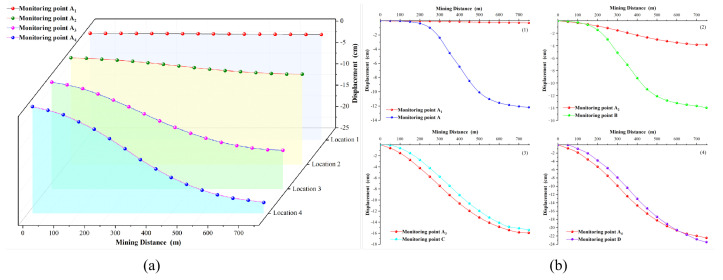
Comparison between the displacement of the corresponding position during mining and the change of the displacement of the top of the mountain: (**a**) variation of mountaintop displacement; (**b**) comparison of displacement changes.

**Figure 11 sensors-22-07811-f011:**
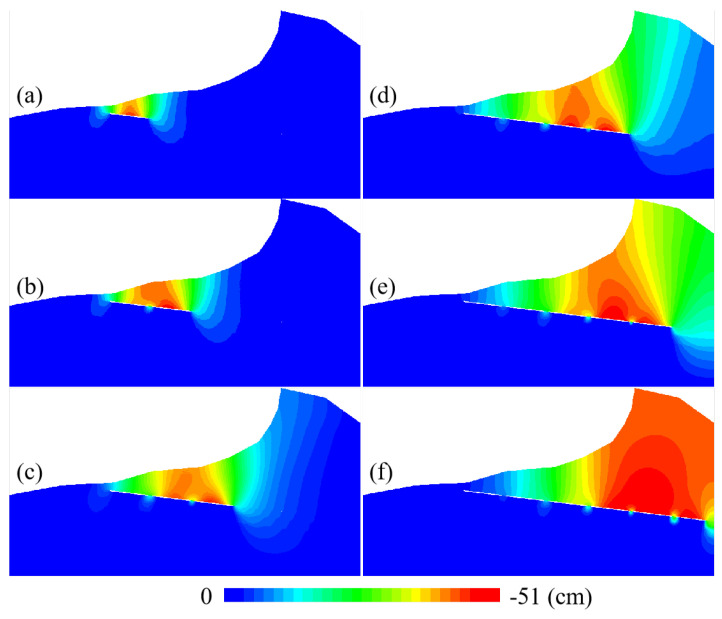
Variation of vertical displacement: (**a**) simulation of displacement change after the 1st mining; (**b**) simulation of displacement change after the 2nd mining; (**c**) simulation of displacement change after the 3rd mining; (**d**) simulation of displacement change after the 4th mining; (**e**) simulation of displacement change after the 5th mining; (**f**) simulation of displacement change after the 6th mining.

**Figure 12 sensors-22-07811-f012:**
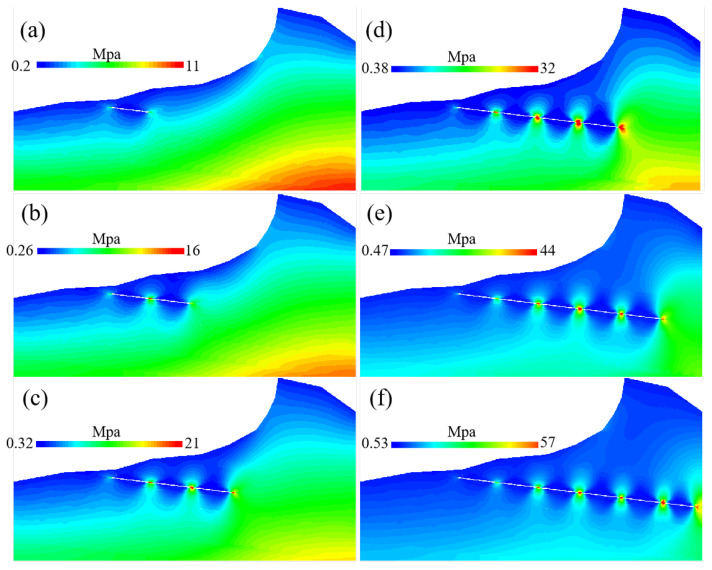
Vertical stress distribution: (**a**) simulation of stress change after the 1st mining; (**b**) simulation of stress change after the 2nd mining; (**c**) simulation of stress change after the 3rd mining; (**d**) simulation of stress change after the 4th mining; (**e**) simulation of stress change after the 5th mining; (**f**) simulation of stress change after the 6th mining.

**Figure 13 sensors-22-07811-f013:**
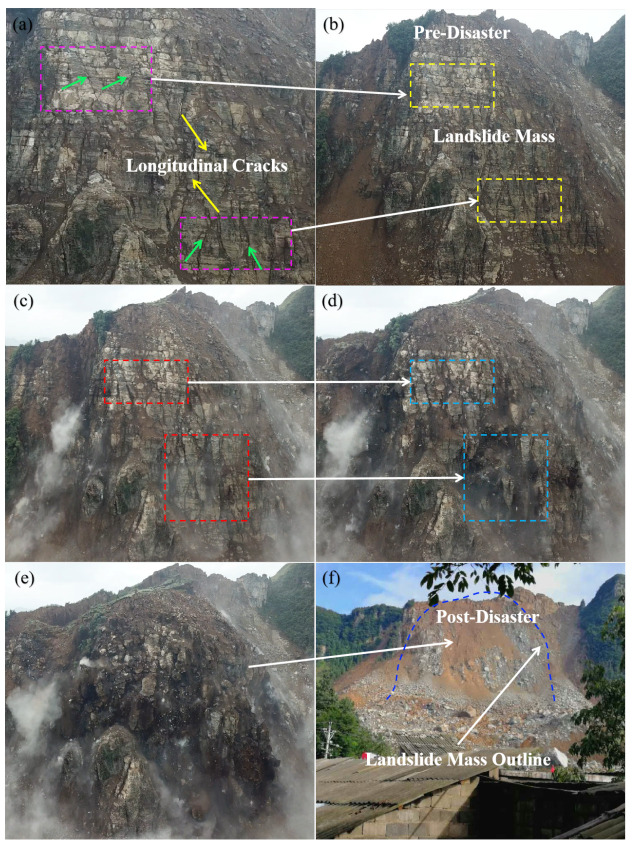
Landslide process: (**a**) the surface fissures of DCB; (**b**) fissures developed on the surface of DCB before the disaster; (**c**) fissures when DCB began to collapse; (**d**) fissures that develop as the DCB collapses; (**e**) collapsing DCB; (**f**) post-disaster appearance.

**Table 1 sensors-22-07811-t001:** Mining location and schedule.

Location	M14,M16 Area	M14,M16 Area	11,008Working Face	11,010Working Face	11,013Working Face
**Time**	2015-01∼2015-09	2015-10∼2016-07	2016-08∼2016-10	2016-12∼2017-02	2017-03∼2017-08
**Closer**	365 m	171 m	94 m	53 m	75 m
**Farther**	578 m	387 m	279 m	155 m	334 m
**Order**	Stage1	Stage2	Stage3	Stage4	Stage5

## Data Availability

Not applicable.

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
