# Peer review of "Instability Monitoring and Numerical Analysis of Typical Coal Mines in Southwest China Based on DS-InSAR"

_sensors, 2022, doi:10.3390/s22207811_

Round 1
Reviewer 1 Report
The paper demonstrates a sound technical and scientific basis and presents a potentially useful numerical approach for assessing the possible instability of a coal mining area, considering the impact of ongoing mining activities.
Nevertheless, the text requires thorough editing to correct the extensive English grammar errors present.
Also, it would be useful to assess the robustness of the numerical model in so far as how it can cope with different ground conditions in various mining locations outside of the case study example(s).
Author Response
Dear Professor,
I genuinely appreciate you taking the time out of your busy day to read this post for me. I sincerely appreciate it. Your feedback on this essay will help me greatly in improving it. At the same time, I sincerely apologize for any inconvenience my poor language, sentence structure, symbolism, etc. may have caused. Every point you made in the article was carefully edited and is now as follows.
We regret the trouble this has caused.
Best wishes.
Comments and Suggestions
The paper demonstrates a sound technical and scientific basis and presents a potentially useful numerical approach for assessing the possible instability of a coal mining area, considering the impact of ongoing mining activities.
Comment 1. Nevertheless, the text requires thorough editing to correct the extensive English grammar errors present.
Comment 2. Also, it would be useful to assess the robustness of the numerical model in so far as how it can cope with different ground conditions in various mining locations outside of the case study example(s).
Response:
Thanks for pointing out the mistake. There are grammatical problems with the article you said. I have re-examined the article thoroughly and corrected the errors. In addition, you said that the robustness of the research methods should be supplemented, and I also explained the robustness of the methods of section 4.3. Because there is no uniform standard for robustness evaluation, I described the method used in this paper, and finally explained the robustness of this research method.

Reviewer 2 Report
In the abstract (line 14), what does "convinced" rules mean?
Line 30: should that be "the country's" rather than "my country's"?
Line 32: "landslides will occur." Is it really a given, that landslides will occur, when there is mining? Or is it "only" an increased risk?
Line 70: please denote the ZSL abbreviation at its first usage.
Line 108: please introduce the TOPS abbreviation.
Line 109: "2.7 3.5 m" is this missing a character or word, like "2.7 - 3.5 m" or "2.7 to 3.5 m"?
Line 124: please introduce the StaMPS abbreviation.
Line 134: please introduce the FaSHPS abbreviation.
It would also be nice to detail the actual methods used, and not leave this as an ambiguous open ended list with "etc".
Line 155: "n-dimensional", the formulas use capital "N", should this be "N-dimensional" instead?
Line 171: Why two diagonal matrices?
Line 187: Is there a reference for the FLAC 3D software?
The description of the employed method is not quite clear and should be better explained in my opinion. This may be also due to some english grammar errors.
Figure 3: I presume that L_1 refers to the blue line, why is a different color chosen for the label? Choosing the same color would more clearly indicate a connection between the two.
It would be nice to offer more details on the discretization in the simulation. Overall, the discrete model lacks a clear explanation.
Equations 8-10 and describing text: two different variants for phi are used in the representation. I think it would be nicer to have this denotation uniformly.
The connection between the sensoric measurement an the numerical simulation could be explained more clearly. The connection is only described fairly broadly. Are there more concrete quantitative measures that are compared between the two?
The rock mass damage degree K is declared as highly important, but is not nearer defined or explained.
Author Response
Dear Professor,
I genuinely appreciate you taking the time out of your busy day to read this post for me. I sincerely appreciate it. Your feedback on this essay will help me greatly in improving it. At the same time, I sincerely apologize for any inconvenience my poor language, sentence structure, symbolism, etc. may have caused. Every point you made in the article was carefully edited and is now as follows.
We regret the trouble this has caused.
Best wishes.
Comments and Suggestions
Comment 1. In the abstract (line 14), what does "convinced" rules mean?
Response: Thanks for pointing out the mistake. I'm sorry, what I'm trying to say is that the experimental results show certain rules. This is my mischoice of words. I have changed "convinced " to "certain ".
Comment 2. Line 30: should that be "the country's" rather than "my country's"?
Response: I'm sorry that I didn't make myself clear here. I have revised it to: Southwest China is also one of China's major coal mining areas
Comment 3. Line 32: "landslides will occur." Is it really a given, that landslides will occur, when there is mining? Or is it "only" an increased risk?
Response: I quite agree with you. Landslides may not happen, but will only increase the risk of landslides. I have revised it according to your suggestion.
Comment 4. Line 70: please denote the ZSL abbreviation at its first usage.
Response:Sorry, I just wanted to make the article look concise, so I abbreviated some place names, but changed them according to your suggestion.
Comment 5. Line 108: please introduce the TOPS abbreviation.
Response: The introduction of TOPS (Terrain Observation by Progressive Scans) has been added to the article.
Comment 6. Line 109: "2.7 3.5 m" is this missing a character or word, like "2.7 - 3.5 m" or "2.7 to 3.5 m"?
Response: I'm sorry, there is a character missing here, like "2.7~3.5 m", which is an undue mistake. I have revised it in the article.
Comment 7. Line 124: please introduce the StaMPS abbreviation.
Response: The introduction of StaMPS (Stanford Method for Persistent Scatterers) has been added to the article.
Comment 8. Line 134: please introduce the FaSHPS abbreviation.
Response: The introduction of FaSHPS(Fast Statistically Homogeneous Pixel Selection) has been added to the article.
Comment 9. It would also be nice to detail the actual methods used, and not leave this as an ambiguous open ended list with "etc".
Response: Thank you for your opinion. I have revised it according to your opinion and deleted "etc" at the end.
Comment 10. Line 155: "n-dimensional", the formulas use capital "N", should this be "N-dimensional" instead?
Response: Thank you so much for checking my article so carefully. I have revised it according to your suggestion. Here, "n-dimensional" is changed to "N- dimensional".
Comment 11. Line 171: Why two diagonal matrices?
Response: After decomposition, SCM is actually two matrices, a diagonal matrix ,, and a full-rank real symmetric matrix ,, .The two complex diagonal matrices referred to in this paper are , and .
Comment 12. Line 187: Is there a reference for the FLAC 3D software?
Response: I have consulted a lot of materials and found that there are few related documents or the reference value is not very high. Because my major is mining engineering, FLAC 3D is often used for engineering analysis of mining. This paper is based on my major and DS-InSAR technology, and no reference is attached.
Comment 13. The description of the employed method is not quite clear and should be better explained in my opinion. This may be also due to some english grammar errors.
Response: I'm very sorry, but due to some grammatical errors, the method adopted has not been clearly stated. I have checked and corrected the grammar of this article. Sorry for the inconvenience caused to you.
Comment 14. Figure 3: I presume that L_1 refers to the blue line, why is a different color chosen for the label? Choosing the same color would more clearly indicate a connection between the two.
Response: I'm sorry, I didn't think about this problem carefully. According your view, in figure 3, I have changed the color of the label to the same color as L_1.
Comment 15. It would be nice to offer more details on the discretization in the simulation. Overall, the discrete model lacks a clear explanation.
Response: Dear Professor, it may be the inconvenience caused by unclear expression. I have completely modified this article. The main purpose of numerical simulation is to analyse the force distribution, displacement change and mining influence range of mountain under the condition of coal mining. The details of numerical simulation are described in the discussion section. I didn't understand what you mean by discretization of models. The FLAC3D model is a three-dimensional finite difference model, belonging to a continuous model.
Comment 16. Equations 8-10 and describing text: two different variants for phi are used in the representation. I think it would be nicer to have this denotation uniformly.
Response: I have revised it in the article according to your advice.
Comment 17. The connection between the sensoric measurement an the numerical simulation could be explained more clearly. The connection is only described fairly broadly. Are there more concrete quantitative measures that are compared between the two?
Response: I'm very sorry, maybe I didn't clearly describe the relationship between the two methods in the text, which caused you inconvenience. In the discussion part, the relationship between InSAR technology and numerical simulation, model building and mountain stability analysis can reflect their relationship. And in section 4.3, I re-summarized the relationship between the two.
Comment 18. The rock mass damage degree K is declared as highly important, but is not nearer defined or explained.
Response: I'm sorry, I thought this was not the point of the article, so I just mentioned it casually. However, according to your opinion, I added the definition of K.

Reviewer 3 Report
In this paper, the remote sensing technology and numerical method are combined to study the stability of the mountain in mining area. At present, there are few similar studies. Combining the two methods to analyze the stability of mountain is the innovation of this paper, and the method proposed is feasible. However, there are some inappropriate mistakes in this article, which need to be improved seriously.
1. Abbreviations such as "ZSL", "HQ" and "TOPS" are not introduced in many places in the article.
2. In the part of discussion, the relationship between remote sensing methods and numerical analysis methods is not clear.
Author Response
Dear Professor,
I genuinely appreciate you taking the time out of your busy day to read this post for me. I sincerely appreciate it. Your feedback on this essay will help me greatly in improving it. At the same time, I sincerely apologize for any inconvenience my poor language, sentence structure, symbolism, etc. may have caused. Every point you made in the article was carefully edited and is now as follows.
We regret the trouble this has caused.
Best wishes.
Comments and Suggestions
In this paper, the remote sensing technology and numerical method are combined to study the stability of the mountain in mining area. At present, there are few similar studies. Combining the two methods to analyze the stability of mountain is the innovation of this paper, and the method proposed is feasible. However, there are some inappropriate mistakes in this article, which need to be improved seriously.
Comment 1. Abbreviations such as "ZSL", "HQ" and "TOPS" are not introduced in many places in the article.
Response: I'm sorry, but I've corrected these mistakes in the article.
Comment 2. In the part of discussion, the relationship between remote sensing methods and numerical analysis methods is not clear.
Response: I'm sorry, the previous description of the relationship between DS-InSAR and numerical simulation is somewhat vague. I have already described the relationship between these two methods in section 4.3.

Round 2
Reviewer 2 Report
Dear authors,
thanks a lot for revising and improving your manuscript. I think it is much better readable now. I only stumbled on minor points, detailed below.
Line 32: missing space after comma
Line 33: missing space after comma
Line 157: missing space after comma
I think the explanation of the symbol phi in equation 3 is missing.
Line 184: missing space after comma
I still think that a reference to the software (https://www.itascacg.com/software/flac3d) would be helpful to clarify which software specifically you are using.
Author Response
Dear Professor,
Thank you from the bottom of my heart for checking my manuscript again in time, and pointing out many mistakes caused by my carelessness. Thank you very much for your patience. All the mistakes you pointed out have been corrected this time.
Best wishes.
Comments and Suggestions
Comment_1. Line 32, Line 33, Line 157, Line 184: missing space after comma.
Response: Thanks for pointing out these mistakes. I've added a space in the corresponding position.
Comment_2. I think the explanation of the symbol phi in equation 3 is missing.
Response:I'm terribly sorry, I was negligent. I have added the explanation of phi in the corresponding place.
Comment_3. I still think that a reference to the software (https://www.itascacg.com/software/flac3d) would be helpful to clarify which software specifically you are using.
Response: After consideration, I agree with your opinion. I've added a reference to line 189.
